# A Textile Sensor for Long Durations of Human Motion Capture

**DOI:** 10.3390/s19102369

**Published:** 2019-05-23

**Authors:** Sufeng Hu, Miaoding Dai, Tianyun Dong, Tao Liu

**Affiliations:** 1State Key Laboratory of Bioelectronics, School of Instrument Science & Engineering, Southeast University, Nanjing 210096, China; 230188148@seu.edu.cn; 2State Key Laboratory of Fluid Power and Mechatronic Systems, School of Mechanical Engineering, Zhejiang University, Hangzhou 310027, China; m.dai@u.northwestern.edu (M.D.); tianyundong@zju.edu.cn (T.D.)

**Keywords:** textile sensor, knee joint, measurements, daily activities

## Abstract

Human posture and movement analysis is important in the areas of rehabilitation, sports medicine, and virtual training. However, the development of sensors with good accuracy, low cost, light weight, and suitability for long durations of human motion capture is still an ongoing issue. In this paper, a new flexible textile sensor for knee joint movement measurements was developed by using ordinary fabrics and conductive yarns. An electrogoniometer was adopted as a standard reference to calibrate the proposed sensor and validate its accuracy. The knee movements of different daily activities were performed to evaluate the performance of the sensor. The results show that the proposed sensor could be used to monitor knee joint motion in everyday life with acceptable accuracy.

## 1. Introduction

Monitoring human joint movements continuously under daily environments is important for rehabilitation, sports medicine, and virtual training. At present, the perfect methods of joint detection are mainly limited in the laboratory environment. For example, optical motion capture systems, which have high accuracy, are regarded as the golden standard of human motion measurement [1]. However, a series of mark points need to be attached to the human body when using the equipment. In addition, the operation is tedious and the price of the equipment is very expensive. Goniometers, having high accuracy, are also used to measure the movement of human joints [2]. Unfortunately, the equipment is large in size and heavy in weight, and is not suitable for long-term wear.

In the past decade, inertial measurement units (IMUs), which consist of accelerometers, gyroscopes, and magnetometers, have made great progress in wearable measurements [3,4]. However, it is necessary to establish a human motion model to obtain human motion parameters, which is very complex and cumbersome. Moreover, the sensors are usually tied to the body, and relative displacement often occurs during movement, resulting in inaccurate results. Lastly, the sensors are uncomfortable to wear and interfere with daily activities due to their solid structures.

In order to improve the comfortability of sensor interactions with people, flexible sensors based on soft conductive materials have received a lot of attention. For example, conductive carbon nanotubes were filled into polydimethylsiloxane as a stretchable strain sensor to monitor human activities [5]. Silver nanowires were blended in a conductive elastomer as a flexible sensor for human motion detection [6]. Although these sensors are very soft, they have high hysteresis due to the characteristics of the materials. In addition to the comfortability, other requirements for soft wearable sensors are for them to have electrically secure and mechanically reliable characteristics in situations where the system is stretched and/or damaged. Some studies have considered this issue by developing stretchable and self-healing electronics [7,8,9]. However, the fabrication processes of these flexible sensors are complex and takes long times.

In this paper, we designed a new flexible textile sensor, which was made of ordinary fabrics and conductive yarns, with a simple process. A wearable flexible textile sensor system was developed to capture knee joint movements. Tensile tests were performed to evaluate the properties of the sensor. Experiments of knee movements indoors and outdoors were conducted to evaluate the performance of the sensor.

## 2. Method

### 2.1. Structure and Materials

Figure 1 shows the structure of our textile sensor. The sensor had two layers (an upper layer and a lower layer), as shown in Figure 1a. Both layers were made of elastic fabric and inelastic fabric. Commercially available elastic bands were used as the material of the elastic fabric, and denim were used as the material of the inelastic fabric. The two layers were placed in opposite directions. Then, smooth silk was used to wrap the two layers, in order for them to make full contact while limiting the friction.

Figure 1b shows the structure of the upper layer. A conductive yarn was sewn in a V shape into the inelastic fabric of the upper layer. A kind of silver-coated yarn named “Agposs T1” (purchased from Mitsufuji Corporation, Kyoto, Japan), which was washable and wear-resistant with uniform conductivity, was used as the material of the conductive yarn. 

Figure 1c shows the structure of the lower layer. The conductive yarn was sewn in a straight line into the inelastic fabric of the lower layer. An electrode marked ‘B’ was at the end of the inelastic fabric. On the other side of the lower layer, an electrode marked as ‘C’ was sewn at the junction of the two fabrics. The smooth silk was fixed at electrode C. 

Figure 1d shows a picture of the flexible textile sensor. It can be found that the sensor is very small, light-weight, and can be easily imbedded to our clothes for wearable measurement.

### 2.2. Working Principle

The sensor works like a sliding rheostat, as shown in Figure 2a,b. The conductive yarn was used to densely sew electrode C such that the resistance of electrode C is negligible compared to the conductive yarn. When the sensor is stretched, the resistance between electrode A and electrode C changes with the displacement.

Assuming that the resistance change of the sensor $\Delta R_{x}$ is proportional to its change of length Δl, the sensitivity is written as:(1)$$S_{0}=\frac{\Delta R_{x}}{\Delta l}.$$

However, since the displacement of the human joint is not particularly large, the resistance change is small. In order to improve the resolution of the sensor, the V-shaped conductive yarns were used. The length of electrode C in the stretching direction must be greater than the distance between the two adjacent V-shaped conductive yarns in the stretching direction to prevent electrode C from coming out of contact with the V-shaped yarns during the stretching process, which would result in circuit breaking. During the relative motion of electrode C and the V-shaped conductive yarns according to Figure 2a,b, the resistance between electrode A and electrode C makes a sudden change. Figure 2a shows that the resistance between electrodes A and C is *Rv*, and the resistance stays constant at a distance during the movement of electrode C. When electrode C is separated from the M point (midpoint of one side of the V-shaped yarns), the resistance between electrodes A and C immediately becomes 2 *Rv*. According to the working principle, the sensor has certain nonlinearity characteristics, which can be modified by designing a more dense V shape.

For the knee motion measurements, two sides of the sensor are fixed onto the thigh and shank, respectively, as shown in Figure 2. When the knee is bent, the sensor will be stretched by the two fixed sides. The change in length of the inelastic fabric can be neglected compared with that of the elastic fabric. Electrode C moves along the inelastic fabric and makes contact with the conductive yarn of the upper layer in different positions, resulting in the change of resistance between electrodes A and B.

In this paper, we consider the knee as a single radial model [10]. The relationship between Δl and the angle change of the knee joint Δθ can be illustrated by
(2)Δl=r·Δθ
where *r* is the radius of the knee joint. 

By combining Equations (1) and (2), Δθ can be determined by the resistance change ΔRx of the sensor:(3)$$\Delta \theta =\frac{1}{r\;\cdot \;S_{0}}\cdot \Delta R_{x}.$$

## 3. Experimental

The sensor was fixed onto the thigh and calf of a subject by straps. The subject had given his informed consent for inclusion before he participated in the study. The study was approved by the Medical Ethics Committee of School of Medicine, Zhejiang University. Data were collected by a slave computer and sent to the host computer through wireless technologies for further monitoring.

### 3.1. Sensor Performance Test

In order to determine the maximum displacement required by the sensor, we used a soft ruler to measure the change of knee joint angle of 100° (from 180 to 80°). We found that the maximum displacement required by the sensor was 50 mm, and we used this value to design the sensor. The tensile tests were conducted on a universal testing machine (Model QX-W200, Shanghai, China) at a speed of 10 mm/min to evaluate the performance of the sensor. Both stretching and contraction tests were performed repeatedly, in which the elongation and resistance of the sensor were recorded.

### 3.2. Calibration

Calibration is needed before the use of the sensor. A standard measuring device, an electrogoniometer (RE), was used as the reference to calibrate the sensor, as shown in Figure 3. The RE consisted of two planks and one potentiometer. 

The calibration process was as follows: The sensor and RE were both placed onto the thigh and shank of the subject. The calibration procedure was started after the successful connection of the slave computer and the host computer, which was done automatically after the subject finished knee movements as required.

### 3.3. Knee Movement Indoors

The subject was asked to sit on the ground and perform knee extension and flexion movements. A six-camera optical motion capture system, VICON (T40s, Oxford, UK), acted as the gold standard for measurement. A wearable sensing system was developed to capture knee joint movements. The data of RE and the flexible fabric sensor were recorded simultaneously with the VICON.

### 3.4. Sensor Performance Test of Knee Movement of Daily Life Outdoors

The subject performed trials of daily movements in each of the following conditions: Up and down stairs, walking normally on level ground, and standing up and sitting down on level ground. These actions were completed in succession and each action lasted 10–15 s. The results of the sensor were evaluated through comparison with those of the RE.

## 4. Results and Discussion

### 4.1. Tensile Testing of the Flexible Textile Sensor

The results of tensile testing are shown in Figure 4. The curves of the stretching test and the contraction test are not completely linear, and there are some fluctuations which are decided by the design principle of the sensor. Two linear functions are used to fit the two curves.

For the stretching test, the fitting linear function is:*y* = 0.9629*x* + 64.419(4)

For the contraction test, the fitting linear function is:*y* = 0.9582*x* + 65.258(5)
where *x* is the elongation distance and *y* is the resistance of the sensor. 

The coefficients of determination for the linear regression curve are 0.9838 and 0.9815 for the stretching test and the contraction test, respectively. The sensitivities of the sensor are 0.9629 and 0.9582 Ω/mm for the stretching test and the contraction test, respectively. 

The results show no hysteresis and validate the reproducibility of the sensor, which is evidenced by the following daily activity experiments. Compared to wearable sensors for capturing joint movement [2,11,12,13,14,15], the same functionality is achieved by the sensor with a simple and novel structure in this paper. In most studies, conductive materials in the sensor are directly involved in deformation [12,13,14]. The conductivity of the material is prone to be affected by the relaxation and fatigue of the material due to frequent deformation, which would lead to hysteresis and poor durability of the sensor [12]. In this paper, deformation and conduction are separated. The elastic fabric of the sensor is used for deformation, and conductive yarns are embedded in the inelastic fabric to avoid deformation. This minimizes the impact of deformation on the conductive yarns, which improves the performance of the sensor and extends its life. 

### 4.2. Knee Movement Indoors

Figure 5 shows the results that the sensor used to detect the motion of the knee joint in the laboratory, and the results were compared with those of the RE and VICON. The results show that the sensor can detect the movement of the knee joint very well, because the knee angle of the sensor has similar trends with those of the RE and VICON. The correlation coefficient is 0.91 between the results of the sensor and VICON. The main error lies in the peak value of the curve, which is relative to the maximum bending and stretching angle of the knee joint movement. However, the peak error is stable, which is evidenced by the sensor’s detected angles, which are basically below those of the RE and VICON at the peaks, and above those of the RE and VICON at the troughs.

### 4.3. Knee Movement of Daily Life Outdoors

The test results of daily activity are shown in Figure 6, and were compared with those of the RE system. The results show that the curve of the sensor has a high correlation coefficient (0.92) with the curve of the RE, which mean that the sensor can detect daily activity well and without hysteresis. 

Continuous monitoring of knee motion under daily environments is necessary for rehabilitation and daily care of patients. However, only a few works provided good results of knee motion under daily conditions [4,16,17]. Our flexible sensor was made of fabric with good accuracy, which can be embedded in garments without affecting the daily activities of humans.

## 5. Conclusions

In this paper, we developed a new flexible textile sensor to measure movements of knee joints using ordinary fabric and conductive yarns. The sensor is small, light, low-cost, and suitable for long-term wear. Good linearity with no hysteresis was shown in the tensile test. The correlation coefficient of the sensor is 0.92 compared with the standard measuring device when monitoring the daily movement of the knee joint. This kind of flexible textile sensor can be integrated into clothing for long durations of noninvasive knee motion monitoring. This paper only provides the motion detection experiment of human knee joints, but the motion of other joints can also be monitored by this sensor design principle.

## Figures and Tables

**Figure 1 sensors-19-02369-f001:**
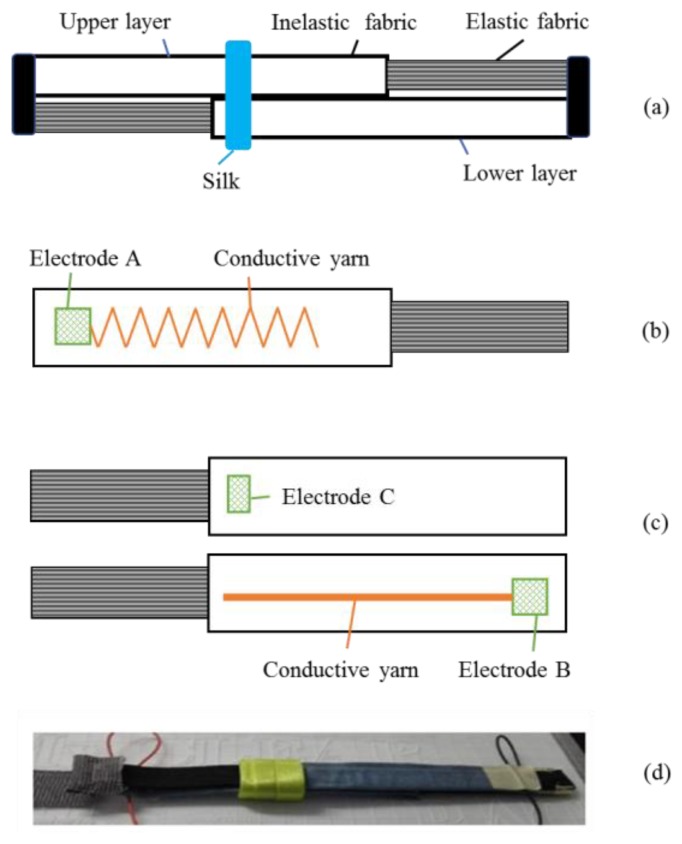
(**a**) The structure of the sensor; (**b**) the upper layer; (**c**) the lower layer; and (**d**) a picture of the sensor.

**Figure 2 sensors-19-02369-f002:**
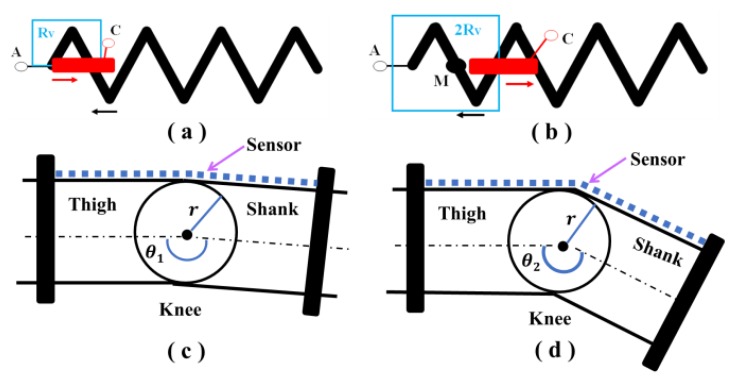
(**a**,**b**) Schematic diagrams of the working principle of the textile sensor. (**c**,**d**) The working principle of the sensor for knee movement measurements.

**Figure 3 sensors-19-02369-f003:**
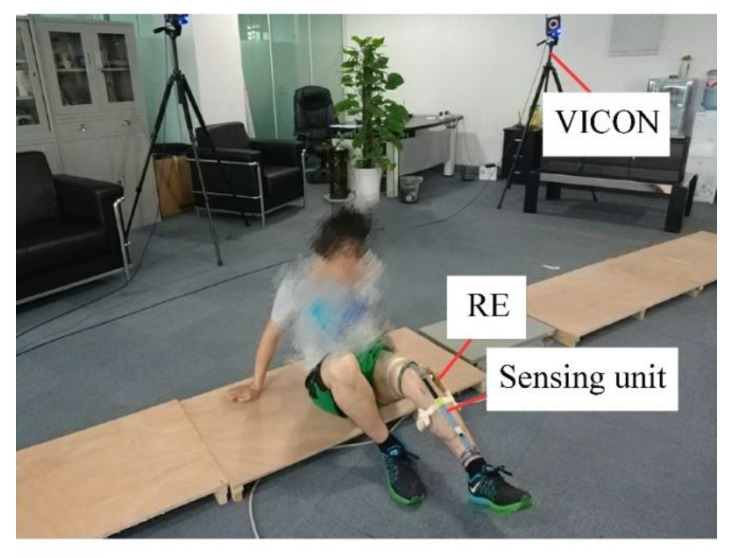
The experiment of knee movement.

**Figure 4 sensors-19-02369-f004:**
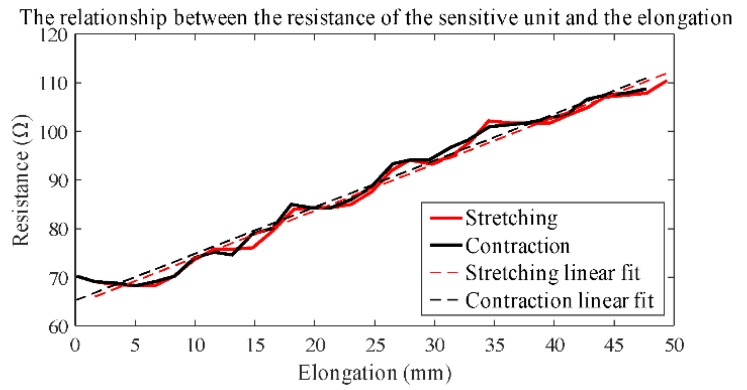
The tensile testing results of the flexible textile sensor.

**Figure 5 sensors-19-02369-f005:**
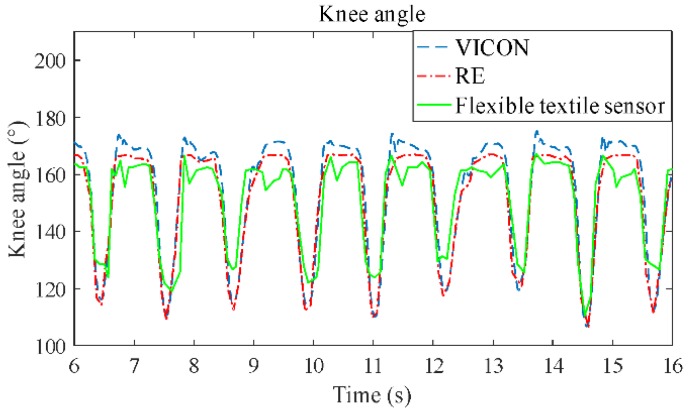
The comparison of angle measurement with the electrogoniometer (RE) and VICON systems, and the flexible textile sensor.

**Figure 6 sensors-19-02369-f006:**
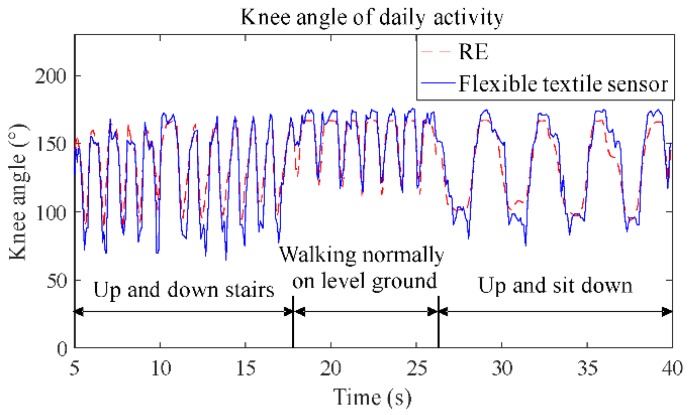
The comparison of angle measurement between the RE system and the flexible textile sensor.

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
