# Peer review of "A Textile Sensor for Long Durations of Human Motion Capture"

_sensors, 2019, doi:10.3390/s19102369_

Round 1
Reviewer 1 Report
Summary Comments:
Sufeng Hu et al. reported on a flexible textile sensor to record human knee movement data for long-term measurement. Their simple structured sensor showed a reliable sensitivity (~96%), comparable to those of conventional sensors and no hysteresis during several common knee movement tests. Their sensors as a kind of rheostat are able to measure resistance in real-time. Their sensors operate with a concept of sensing knee motion through detecting resistance changes when the distance between the electrodes increases as the knee bends. The concept looks interesting.
After implementation, the authors conducted the tensile test for performance evaluation of the sensor. They verified a sensitivity of the sensor by measuring resistance change through stretching-releasing test. They also demonstrated there were no hysteresis on its tensile performances. Additionally, a test subject wore their sensor system on knee to monitor its performance during joint motion. They determined 3 patterns of movement (Up and down stairs, normal walking, up and sit down) and compared those results with their conventional sensors (electrogoniometer and commercial motion capture system). As a result, their flexible textile sensor showed similar sensitivity compared to other sensors (91~92%) and there was no hysteresis on cyclic motion test so they proved their sensor has a good accuracy to monitor knee motion of daily activities.
Taken together, it can be summarized that the authors’ works is worthy as their sensors were proposed with on unique and simple ideas. Based on this study, the reviewer thinks this manuscript should be published after several revisions.
Specific Comments:
Comment #1: In applications of wearable electronics and sensors, it is very important to secure electrically and mechanically reliable characteristics in a situation where the system is stretched and/or damaged. To better clarify this issue, the authors should emphasize stretchable and self-healing electronics in Introduction part. The reviewer suggests the following references.
1. "Multifunctional wearable devices for diagnosis and therapy of movement disorders", Nature Nanotechnology, 2014, 9, 397-404
2. "An integrated self-healable electronic skin system fabricated via dynamic reconstruction of a nanostructured conducting network", Nature Nanotechnology, 2018, 13, 1057-1065
3. "Nanomaterials in Skin-Inspired Electronics: Toward Soft and Robust Skin-like Electronic Nanosystems", ACS Nano, 2018, 12, 11731-11739
Comment#2: In line 154, page 3, the expression “same functionality” does not seem clear. Besides the sensitivity, the performance of the sensor is a comprehensive concept which includes the indices such as accuracy and response time. So the authors need to evaluate other performances to clarify the functionality of their sensors.
Comment #3: If the authors want to assert that their sensors are capable of long-term operation, they should verify that the sensor has mechanical durability through about 1000 cycles of repetitive operation. This reviewer is concerned that electrodes or conductive yarn of the sensors may be mechanically degraded by repetitive operations.
Comment #4: The authors' sensors are worn in direct contact with the skin of the legs. Generally, a device with a high modulus causes some side effects such as inflammation and allergies when long-term contact with the skin tissue. But in this text, the authors did not suggest any specific data or objective basis that their sensors have a long-term comfort fit. Therefore, the authors should evaluate the mechanical properties such as Young’s modulus of their devices and objectively evaluate them whether they have a comfortability.
Comment #5: The three types of daily activity tests wearing the sensors performed by the authors in this study do not seem to be sufficient. The reviewer recommends that the authors perform a more various type of daily activity tests, such as kicking a ball, jumping in the ground, doing squats.
Comment #6: In line 89, page 3, the authors introduced the term “M point”, which must be meant to middle point between electrode A and electrode C. but they did not mention any definition of M point. The authors must give a definition of a terminology which they newly introduced. So that should be clarified.
Comment #7: There are several typos, inappropriate expressions, and grammatical errors in the manuscript.
#7-1: in line 13, page 1, the expression in sentence “and comfortable for is suitable for long durations of human motion capture is still an ongoing issue.” is syntactically wrong. The sentence should be re-written correctly.
#7-2: in line 24, page 1, perfected to perfect
#7-3: in line 36, page 1, Last but not least to lastly
#7-4: in line 71, page 3, the figure numbering is wrong. Figure 3 to Figure 2
#7-5: in line 71, page 3, (a, b) to (a) and (b), (c, d) to (c) and (d)
#7-6: in line 73, page 3, (a, b) to (a) and (b)
#7-7: in line 86-87, page 3, the expression in sentence “the resistance between the electrode A and the electrode C have abrupt.” is syntactically wrong. The sentence should contain a noun.
#7-8: in line 91, page 3, It is more appropriate to change the expression "reduced" to "modified".
#7-9: in line 93, page 3, the sentence “the change of the inelastic fabric in length can be neglected compare with the elastic fabric.” is locally wrong. The comparison target of the “change (amount)” should be the “change (amount)”. The expression “the elastic fabric” should be replaced by “that of the elastic fabric.”.
#7-10: in line 131, page 4, Date to Data
#7-11: in line 146, page 5, stretching test to contraction test
#7-12: in line 164, page 5 & Figure 5, page 6, VIOCN to VICON
#7-13: in line 169-170, page 6, RE and VICON and RE to RE and VICON
#7-14: in line 180, page 6, the figure numbering is wrong, Figure 1 to Figure 6
#7-15: in line 180, page 6, VICON to RE
Besides, the reviewer recommends the authors that the manuscript be reviewed entirely to correct typos, grammatical errors, and inappropriate expressions. This manuscript contains many errors in the text.
Author Response
Thanks for your careful and thorough review and for your kind suggestions, we have attempted to satisfy all the concerns through either changes to the manuscript or a detailed explanation. Please find the revised manuscript and a point-by-point reply to the questions. In our view, the revised manuscript is much improved by having these comments. Our point-by-point responses to the concerns are listed as following.
1. In applications of wearable electronics and sensors, it is very important to secure electrically and mechanically reliable characteristics in a situation where the system is stretched and/or damaged. To better clarify this issue, the authors should emphasize stretchable and self-healing electronics in Introduction part. The reviewer suggests the following references.
Reply:
We have added the stretchable and self-healing electronics in introduction part and added three references you recommended.
In addition to the comfortability, another requirement for soft wearable sensors is to secure electrically and mechanically reliable characteristics in a situation where the system is stretched and/or damaged. Some studies have considered this issue by developed the stretchable and self-healing electronics [7-9].
2. In line 154, page 3, the expression “same functionality” does not seem clear. Besides the sensitivity, the performance of the sensor is a comprehensive concept which includes the indices such as accuracy and response time. So the authors need to evaluate other performances to clarify the functionality of their sensors.
Reply:
The “same functionality” in this paper referred to the front of the words “capturing joint movement”. The accuracy of our sensor have compared standard devices VICON and RE. Our sensors can monitor knee joint motion in the daily life, indicating that the response time of the sensor is very fast.
3. If the authors want to assert that their sensors are capable of long-term operation, they should verify that the sensor has mechanical durability through about 1000 cycles of repetitive operation. This reviewer is concerned that electrodes or conductive yarn of the sensors may be mechanically degraded by repetitive operations.
Reply:
Thanks for this suggestion. There is no need to worry about the repeatability of the sensor, because the conductive wires of the sensor are Agposs T1 (purchased from Mitsufuji Corporation, Japan). The Agposs T1 can wash for more than 1000 times. The senor works with only a small amount of friction, so the sensor can be used for a long time (more than 1000 cycles).
4. The authors' sensors are worn in direct contact with the skin of the legs. Generally, a device with a high modulus causes some side effects such as inflammation and allergies when long-term contact with the skin tissue. But in this text, the authors did not suggest any specific data or objective basis that their sensors have a long-term comfort fit. Therefore, the authors should evaluate the mechanical properties such as Young’s modulus of their devices and objectively evaluate them whether they have a comfortability.
Reply:
Thanks for this suggestion. Our sensor is made of the elastic fabric and the inelastic fabric. The elastic fabric is much soft than our clothes, so it won't give high modulus to our body. On other hand the sensor needs not to touch the skin because the sensor can be sewn on the clothes.
5. The three types of daily activity tests wearing the sensors performed by the authors in this study do not seem to be sufficient. The reviewer recommends that the authors perform a more various type of daily activity tests, such as kicking a ball, jumping in the ground, doing squats.
Reply:
Thanks for this suggestion. We think that the results of the sensor for several daily activities recommended by the reviewer to test sensor results can be attributed to response time of the sensor. The frequency of daily activities that we use in the paper such as “up and down stairs, walking normally on level ground” is greater than the daily activities such as “kicking a ball, jumping in the ground, doing squats” the reviewer mentioned, so the sensor still has a high precision in these activities.
6. In line 89, page 3, the authors introduced the term “M point”, which must be meant to middle point between electrode A and electrode C. but they did not mention any definition of M point. The authors must give a definition of a terminology which they newly introduced. So that should be clarified.
Reply:
Thanks for this suggestion, we are sorry for using the term that we have not defined. The term “M point” means the midpoint of the one side of the V-shaped yarns. We have defined in the paper.
7. There are several typos, inappropriate expressions, and grammatical errors in the manuscript.
Reply:
Thank you very much for finding out so many grammatical errors and spelling mistakes. We have made changes based on your comments, and we have also carefully checked the full text.
Reviewer 2 Report
This paper aimed to develop a new flexible textile sensor for knee joint motion analysis. The authors was developed by using ordinary fabrics and conductive yarns for the sensor. They introduced IMU, motion capture system (VICON), and goniometer. There are pros and cons.
Recently there are several convenient IMU using smartphone or marker less motion capture system based on deep learning technique.
Ex.
https://wearnotch.com
http://www.mousemotorlab.org/deeplabcut
The authors concluded that their sensor is small, light, low-cost and suitable for long term wear. But it would be almost same as current IMU sensors. The result showed the good linearity, but there is no evidence for hysteresis. The authors have to show the relationship between joint angle and resistance during stretching and contracting condition.
Minor
Figure 3 -> Figure 2
Figure 1 -> Figure 6
Author Response
Thanks for your careful and thorough review, your kind suggestions and detail comments are of great help for improving our manuscript.
The authors concluded that their sensor is small, light, low-cost and suitable for long term wear. But it would be almost same as current IMU sensors. The result showed the good linearity, but there is no evidence for hysteresis. The authors have to show the relationship between joint angle and resistance during stretching and contracting condition.
Reply:
Thanks for this suggestion. We carefully read the two products recommended by the reviewer. The reviewer is right that the function of our sensor is same with the IMU function. However, IMU has some shortcomings, which we have already explained in the Introduction part, such as complex algorithm, and solid structures for uncomfortable wearing. Our sensor is made of soft fabric, which can be sewn on the clothes. The sensor is comfortable for long time wearing. The working principle of our sensor is very simple. The movement of human body deforms the sensor, which leads to the change of resistance.
We have shown the hysteresis in Figure 4. The solid line in red is the stretching of the sensor, and the solid line in black is the contraction of the sensor. The relationship between joint angle and resistance can find in Figure 5 and Figure 6. For example, in Figure 5, there are many repeating units, each unit representing a process of stretching and contraction of the sensor. The rising curve indicates the relationship between joint angle and resistance during stretching, and the falling curve indicates the relationship between joint angle and resistance during contracting.
Round 2
Reviewer 2 Report
The document is well revised.